# Puckered and JNK signaling in pioneer neurons coordinates the motor activity of the *Drosophila* embryo

Katerina Karkali [1,2] ✉, Samuel W. Vernon[3,4], Richard A. Baines [3], George Panayotou[2] & Enrique Martín-Blanco [1] ✉

Central nervous system organogenesis is a complex process that obeys precise architectural rules. The impact that nervous system architecture may have on its functionality remains, however, relatively unexplored. To clarify this problem, we analyze the development of the *Drosophila* embryonic Ventral Nerve Cord (VNC). VNC morphogenesis requires the tight control of Jun kinase (JNK) signaling in a subset of pioneer neurons, exerted in part via a negative feedback loop mediated by the dual specificity phosphatase Puckered. Here we show that the JNK pathway autonomously regulates neuronal electrophysiological properties without affecting synaptic vesicle transport. Manipulating JNK signaling activity in pioneer neurons during early embryogenesis directly influences their function as organizers of VNC architecture and, moreover, uncovers a role in the coordination of the embryonic motor circuitry that is required for hatching. Together, our data reveal critical links, mediated by the control of the JNK signaling cascade by Puckered, between the structural organization of the VNC and its functional optimization.

The nervous system architecture is the result of interactions between neurons and glia and axon guidance decisions. However, the mechanism by which the nervous system maintains a mechanically balanced organization, while establishing functional neuronal connectivity during morphogenesis, is not well understood. There is a direct relationship between the 3D architecture of the nervous system and its functionality, as evidenced by the following: 1) the spatial distribution of functional circuits in the cortex of vertebrates strongly overlaps with large-scale microstructure, connectivity, and gene expression patterns[1], 2) neural networks likely minimize their structural costs following the wiring optimization principle proposed by Ramón y Cajal[2]. These costs likely arise from metabolic requirements, signal delay and attenuation, and complex guidance processes[3] that increase with interneuronal distance[4], and 3) the spatial organization of cortical areas appears to be influenced by evolution[5].

We recently showed that the 3D architecture of the *Drosophila* embryonic Ventral Nerve Cord (VNC) is directed by the mechanical coordination of neurons and glia[6]. Further, we have also uncovered that a precise modulation of the activity of the JNK signaling cascade in a subset of early specified neurons is required for proper VNC architectural organization and condensation[7]. This current study aims to determine the importance of an accurate nervous system spatial organization for functionality. We investigate whether JNK-mediated VNC architecture is critical for the emergence of coordinated embryonic motor circuit activity, which mediates larval hatching[8].

JNK signaling cascade can impact neural function via dual routes, the regulation of the transcription machinery (via AP1) and the phosphorylation of cytoplasmic targets in the soma[9,10]. Its best-known role is to coordinate the induction of protective genes in response to

[1]Instituto de Biología Molecular de Barcelona (CSIC), Parc Cientific de Barcelona, Baldiri Reixac 10-12, 08028 Barcelona, Spain. [2]BSRC "Alexander Fleming", 34 Fleming Street, 16672 Vari, Greece. [3]Division of Neuroscience, School of Biological Sciences, Faculty of Biology, Medicine and Health, University of Manchester, Manchester Academic Health Science Centre, Manchester M13 9PL, UK. [4]Present address: Brain Mind Institute, EPFL – Swiss Federal Institute of Technology, VD 1015 Lausanne, Switzerland. ✉e-mail: kkabmc@ibmb.csic.es; embbmc@ibmb.csic.es

oxidative (ROS) stress[9] and axonal regeneration[11]. JNK has also been shown to mediate additional functions, both in *Drosophila* and mice, including axonal growth[12], organelle and protein transport[9,10,13–15], synaptic growth and strength[16], dendrite pruning[17], self-renewal of neuronal precursors[18], neuronal migration[19], glia remodeling[20], and neuroplasticity[21]. We now hypothesize that the JNK pathway may also have a share in optimizing neural function, associated to its regulatory role on the structural organization of the VNC in the developing embryo.

During VNC condensation in *Drosophila* embryos, sporadic muscle contractions initially do not correlate with sensory input[22,23]. However, over time, these contractions evolve into coordinated peristaltic patterns that are dependent on the precise coordination of segmental muscle contractions along the antero-posterior (AP) axis[8]. Here, we show that the JNK pathway plays an essential role in enabling embryonic coordinated motor activity. In the early stages of neuropile formation, the tight control of JNK signaling in a subset of pioneer neurons is crucial for the proper organization of the axonal scaffold and for the native dendrite and axon conformation. These processes are essential for the development of the complex muscle activities leading to hatching behavior. At late stages, the JNK activity is engaged in fine-tuning neurons' soma location, dendritic morphology, and number. Axonal transport of synaptic vesicles in pioneer motoneurons is unaffected by the pathway activity but their resting membrane potential (RMP) is altered as they become hyperpolarized, leading to a considerable reduction of endogenous firing. Our data support a fundamental structural role for the JNK signaling pathway in pioneer neurons that is ultimately linked to embryonic motor pattern optimization, independently of its function in modulating firing capability.

## Results

### The level of JNK activity in pioneer neurons is essential for embryonic motor coordination

Considering the roles of JNK signaling on promoting VNC architectural robustness[7], we investigated the potential functional consequences of releasing the negative feedback control of JNK activity implemented by Puc. We first focused on motor circuits and evaluated late embryonic motility patterns[8,22] (Fig. 1a and b) (see Methods and Supplementary Movie 1). In wildtype conditions, developing embryos execute forward and backward waves of peristalsis by coordinating bilaterally symmetrical muscle contractions[8,24] that progress through five defined maturation stages of precise duration[8] (see also Fig. 1c): muscles initiate their activation pattern delivering brief, isolated, unilateral twitches and sporadic head pinching (stage A), this is followed by a prolonged rest period, occasionally interrupted by uncoordinated single contractions (B). Then, an active stage showing frequent incomplete and complete peristalsis arises (C). This active peristalsis stage precedes a short resting period (D), finally leading to vigorous head movements and multiple complete peristaltic events (E) that result in the rupture of the vitelline membrane and larvae hatching (see also Supplementary Fig. 1 and Supplementary Movie 2). In the complete absence of *puc* (*puc^E69*), when embryos' axonal network is altered and they fail to condense their VNC[7], the progression of motor patterns was interrupted, with embryos remaining indefinitely in a immature stage (A), characterized by occasional unilateral twitches and incomplete peristalsis (Fig. 1d and Supplementary Movie 3). Muscles integrity in this condition was tested by evaluating the expression of Myosin in *puc* mutants. Despite the aberrant shape of *puc* homozygous embryos, consequence of failing to complete dorsal closure, the full set of embryonic muscles was unambiguous (Supplementary Fig. 2).

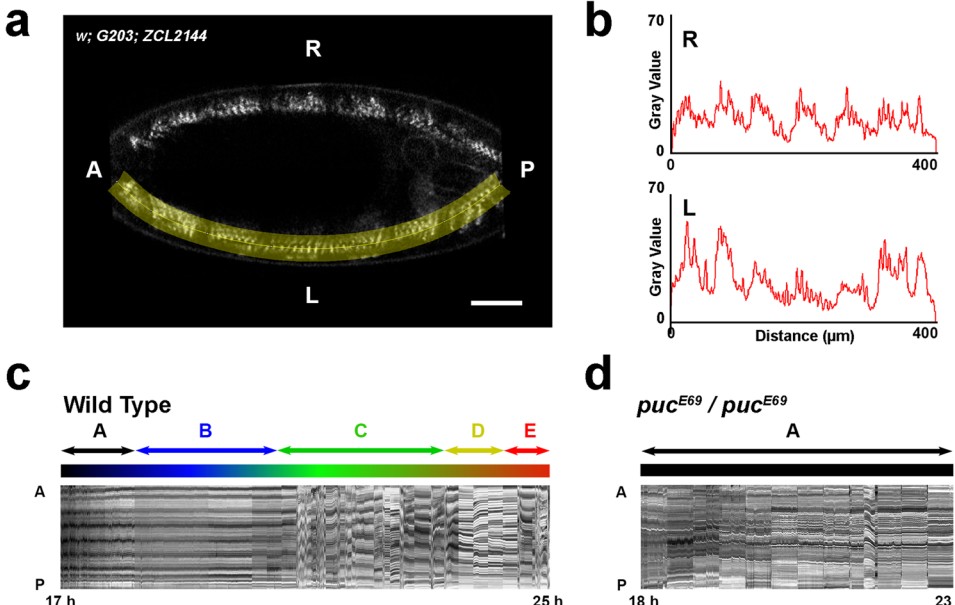

**Fig. 1 | Defective motor coordination results in incomplete peristalsis in *puc^E69* embryos. a** Mid-plane image of a stage 17 representative embryo (*n* = 10) carrying two GFP protein traps expressed at muscle Z-lines (*w; G203; ZCL2144*) (see Supplementary Movie 1). A selected segmented-line (highlighted in yellow) was employed to create kymographs spanning muscle units along the A/P axis (see below) at both sides of the embryo, right (R) and left (L). The evolution overtime of these kymographs was monitored live. Scale bar is 25 μm. **b** Intensity Profile plots generated along the segmented-line selection in (**a**). Gray values as a function of distance highlight the distribution of the segmental muscle units along the axis. Muscle contraction is detected by the differences in the distance between consecutive peaks at sequential time points. **c** Kymograph displaying muscle profiles from 17 to 25 hours AEL for the left side of the embryo in (**a**). Anterior (A) and posterior (P). Motor coordination maturation occurs in 5 stages of approximated stereotyped length[8] (color coded): A, Intense unilateral twitching; B, general inactivity with occasional uncoordinated contractions; C, initiation of backward and forward peristalsis; D, pre-hatching inactivity period and E, complete peristalsis and specialized hatching head movements. **d** Representative kymograph displaying muscle profiles from 18 to 23 hours AEL, when the embryo dies, for the left side of a *puc^E69* embryo (see Supplementary Movie 3). Anterior (A) and posterior (P). Motor coordination fails and embryos do not progress beyond the unilateral contraction period (stage A) (*n* = 13).

The systemic failure of motor coordination observed in *puc* mutants could be assigned to an increase in JNK activity levels in the central nervous system (CNS), but it may also be explained by other undefined non-autonomous events. To rule out the latter possibility, we mimicked *puc* loss of function in different subsets of *puc*-expressing neurons such as the aCC, pCC, RP2 pioneers (using RN2-Gal4) and the VUM midline neurons (using MzVUM-Gal4) by overexpressing a constitutively active form of JNKK (Hep[CA]). It is important to note that interfering with JNK activity in these cells results in architectural defects and the arrest of VNC condensation[7]. An excess of JNK activity in RN2 cells (Fig. 2b and c) or MzVUM cells (Fig. 2d and e) resulted in a phenotype similar to that observed in *puc[E69]* mutant embryos, characterized by occasional unilateral twitches, head pinches, and infrequent incomplete peristalsis (arrested stage A). The embryonic muscles were unaffected ruling out non-autonomous effects (Fig. 2a and b).

To corroborate if the motor defects observed following overexpression of Hep[CA] in RN2 and MzVum cells were specific for these *puc*+ neurons, we employed the CQ2-Gal4, which drives the expression of UAS-linked genes in the U1 to U5 motor neurons, a subset of Eve-positive neurons[25], which also express *puc*[7]. Neither the inhibition, nor the hyperactivation of JNK pathway resulted in any major alteration of either VNC architecture or condensation[7]. We evaluated the progression of motor patterns upon Hep[CA] overexpression in CQ2 cells and found they progressed normally up to the active period C, prolonging indefinitely. These embryos never underwent the last head movements and peristaltic events (period E) needed for hatching (Supplementary Fig. 3). The function of the JNK pathway in CQ2 cells was thus limited to modulating hatching behavior. In short, the arrest in motor pattern maturation observed in *puc* appears to be the consequence of the specific loss of Puc activity in the RN2 and MzVum *puc*+ neurons.

To explore the mechanisms involved in mediating the role of *puc* on motor coordination we focused on the RN2 cells, employing the CQ2-Gal4 targeted neurons as a control. First, we tested if the

downregulation of the JNK pathway may have any effect on the coordination of muscles activities. Second, we evaluated whether interfering in pioneers' neurotransmission may have consequences, either on the structural organization of the VNC or on the coordination of the embryonic motor patterns.

Loss of JNK signaling by overexpression of Bsk[DN] in RN2 (and also in MzVum) cells results in correlation defects in the axonal network, but does not lead to major alterations in VNC condensation[7]. In RN2 cells, the loss of JNK activity just affects the pre-hatching stage (E), which is lengthened and occasionally abortive, with some embryos eventually hatching. Yet, coordinated peristalsis were frequently observed during the stage C and the extended stage E, indicating that muscles coordination was successful (Supplementary Fig. 4a).

Previous reports have shown that pan-neural overexpression of tetanus toxin (TNT), which results in the loss of neurotransmitter exocytosis, does not overly affect embryonic CNS morphology, but leads to loss of muscle contractions. These pseudo-paralyzed embryos fail to hatch[26], while the aCC / RP2 motoneurons show abnormal electrical properties; their depolarization results in significantly more action potentials fired, consistent with a change on their homeostatic setpoint[27]. We found that overexpressing TNT in RN2 cells did not affect the 3D structural architecture of the VNC and mildly its condensation. These embryos showed normal muscle activity coordination (Fig. 3) and progressed up to the resting stage D. This is prolonged indefinitely, and the embryos never undergo the head movements and peristaltic events preceding hatching. Indeed, they fail to hatch. Expression of TNT in CQ2 motoneurons led to equivalent phenotypes; no effect upon VNC's structural organization, or condensation, neither upon muscle activity pattern up to stage D, again, impeding hatching (Fig. 3a–f). It can be concluded that the late behavioral defects caused by the loss of neurotransmitter exocytosis on aCC/RP2 or CQ2 motoneurons are unrelated to the early motor coordination defects observed as a consequence of altering JNK signaling activity in the same cells.

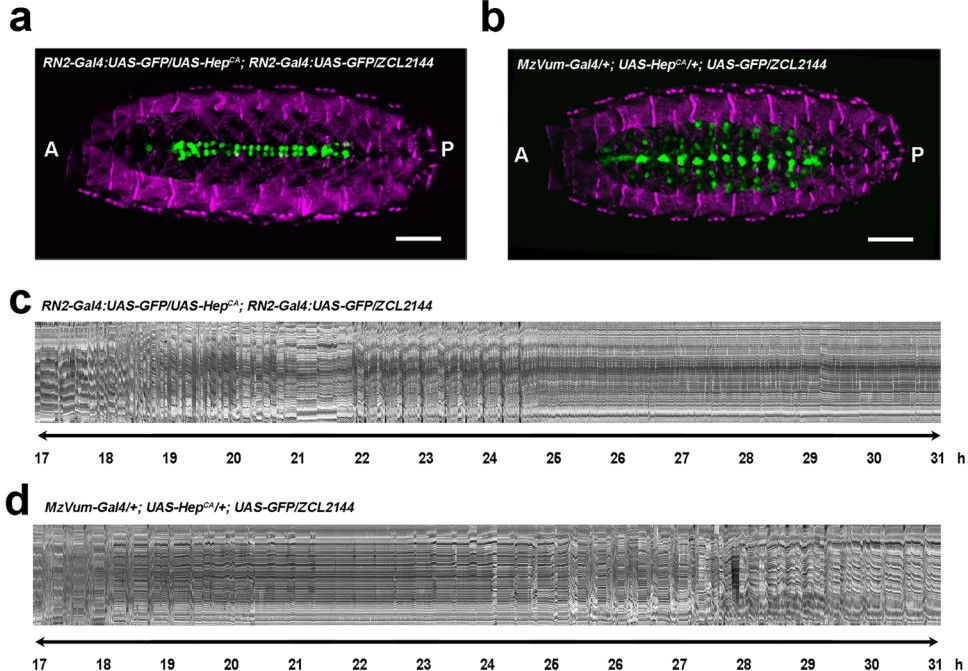

**Fig. 2 | JNK activity modulates embryo motor coordination. a** and **b** Ventral view of stage 17 embryos expressing GFP under the control of the RN2-Gal4 (**a**) (*n* = 12) and MzVum-Gal4 (**b**) (*n* = 11) lines co-expressing Hep[CA] and carrying the muscle Z-lines specific GFP protein trap ZCL2144. Muscles are pseudo-colored in magenta and neurons in green. Scale bar is 25 μm. **c** and **d** Representative kymographs (ZCL2144 marker) revealing occasional unilateral contractions in embryos co-expressing Hep[CA] in the RN2-Gal4 (*n* = 7) (**c**) and MzVum-Gal4 (*n* = 5) (**d**) neurons. These embryos never manage to coordinate muscle movements or progress beyond stage A. Left side is shown.

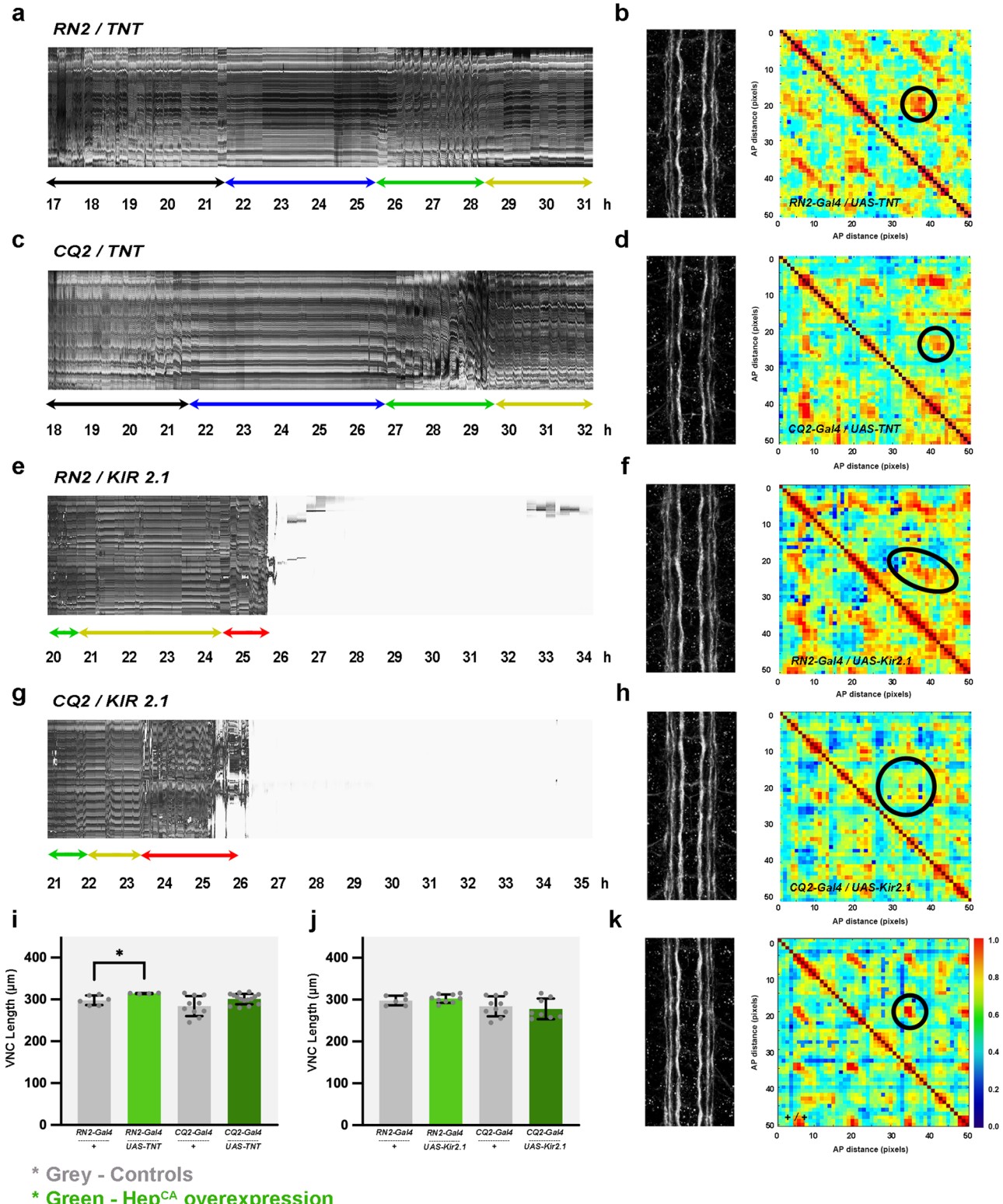

**i** * Grey - Controls

**j** * Green - Hep^CA overexpression

An alternative to interfere in neurons functionality is the overexpression of the human inwardly rectifying K$^+$ channel Kir$_{2.1}$. Expression of Kir$_{2.1}$ hyperpolarizes mammalian neurons[28]. In *Drosophila*, the expression of Kir$_{2.1}$ in aCC results in the almost total absence of excitatory junctional currents in its target muscle due to a failure in the action potential-mediated release of neurotransmitters[27]. We found that the overexpression of Kir$_{2.1}$ in RN2 or CQ2 cells, although it led to an anomalous structural correlation of the VNC, did not affect

VNC condensation or muscles activity and embryos hatched normally. In this condition, the 3D structural nodes developed wider, pointing to a hyperstructured VNC axonal network, different to that observed upon interfering in JNK activity (Fig. 3g–i). Kir$_{2.1}$ overexpression influenced cells dendritic arborization and survival (Supplementary Fig. 5 and Discussion).

The functionality of the TNT and Kir$_{2.1}$ transgenic lines was confirmed by employing a pan-neural driver (ELAV-Gal4). As a

**Fig. 3 | Interference in the physiological function of early pioneers does not affect VNC condensation or embryos motor coordination. a, c, e** and **g** Representative kymographs displaying muscle activity profiles along 14 hours of development for embryos expressing TNT or Kir$_{2.1}$ in RN2 [**a** ($n = 5$) and **e** ($n = 3$)] and CQ2 [**c** ($n = 5$) and **g** ($n = 3$)] neurons. For TNT, motor coordination maturation proceeds to D, the pre-hatching inactivity period, but never reached stage E, complete peristalsis or specialized hatching head movements (**a** and **c**). For Kir$_{2.1}$, no effect was observed on motor coordination and the embryos hatched (**e** and **g**). **b, d, f** and **h** Representative Fas 2 expression profiles and cross-correlation matrices for 17 hours AEL embryos expressing TNT or Kir$_{2.1}$ in RN2 [**b** ($n = 3$) and **f** ($n = 3$)] and CQ2 [**d** ($n = 3$) and **h** ($n = 3$)] neurons. The structural organization of the Fas2 axonal network is largely unaffected for TNT (**b** and **d**). For Kir$_{2.1}$, the structural organization of the Fas2 axonal network was distorted and nodes' sizes and shape (black ellipses) were altered. **i** and **j** Quantification of the VNC length in μm (average and standard deviation) for each condition [TNT - **i** ($n = 6$ for RN2 controls, $n = 4$ for RN2 experimental, $n = 10$ for CQ2 controls and n = 15 for CQ2 experimental) and Kir$_{2.1}$ - **j** ($n = 6$ for RN2 controls, $n = 9$ for RN2 experimental, $n = 10$ for CQ2 controls and $n = 8$ for CQ2 experimental)]. Statistically significant differences in length were detected only in RN2-Gal4<UAS-TNT embryos, using parametric student $t$-tests (*$p = 0.0295$) ($p = 0.2394$ for CQ2-Gal4<UAS-TNT, $p = 0.8877$ for RN2-Gal4<UAS-Kir$_{2.1}$ and $p = 0.7679$ for CQ2-Gal4<UAS-Kir$_{2.1}$). In grey are represented all the control conditions. RN2 and CQ2 conditions are shown in light green and dark green respectively. **k** Control Fas 2 cross-correlation matrix along the AP axis of the VNC of a stage 17 WT embryo stained with Fas 2. The color-coded representation shows the correlation level for each possible comparison at each position along the AP axis. One conspicuous node of robust correlation per segment was detected (black ellipse) ($n = 8$).

consequence of inducing a global neuronal loss of neurotransmiter exocytosis (TNT) or interfering in neuronal polarity (Kir$_{2.1}$), the motor activity of embryos was completely abolished (Supplementary Fig. 4b and c), backing up these lines performance.

Finally, it was previously shown that reduction on the expression of Fasciclin 2 (Fas2), induced by JNK hyperactivation in RN2 neurons, affects axonal network organization, as well as VNC condensation[7]. We then tested whether diminishing Fas2 expression was sufficient for altering motor coordination maturation. Indeed, the motor activities of *fas2*$^{EB112}$ / *fas2*$^{E76}$ trans-heterozygous embryos were completely abolished advocating for the role of JNK activity in pioneer neurons in linking, via Fas2, axonal network organization to motor coordination (Supplementary Fig. 4d).

### JNK signaling does not influence synaptic vesicle transport but modulates motoneuron firing

Would the loss of embryonic motor coordination, after altering JNK activity in *puc*-expressing neurons, be a consequence of the abnormal architectural organization and inability of the VNC to condense? Some *puc+* cells are motoneurons and at anomalous levels of JNK activity, their signaling capability might be impaired, resulting in defects in the embryo motor coordination independently of the VNC's architectural organization. To evaluate this possibility, we studied the axonal transport of synaptic vesicles (measuring Synaptotagmin - Syt) and organelles (Mitochondria - Mito) in the intersegmental nerve [aCC and RP2 (RN2-Gal4) and in VUM motoneurons (MzVum-Gal4)] (see Methods, Fig. 4, Supplementary Fig. 6 and Supplementary Movie 4). Both in RN2 and MzVum axons, synaptotagmin particles moved at a speed of around 0,7 μm/s with a tracking lifetime of 50 s per event. Mitochondria travelled at the same average speed, but their movements were less persistent, remaining on track for an average of 30 s. Particle densities (frequency of events) were equivalent for synaptotagmin and mitochondria, but larger for aCC and RP2 (0,3 N/μm) than for VUMs (0,2 N/μm). No differences were observed between anterograde and retrograde transport. Upon co-expression of Hep$^{CA}$, Synaptotagmin vesicle kinematics were largely unaffected (Supplementary Fig. 6), indicating that microtubule-mediated transport remained fully functional. However, although their average lifetime was sustained, the density of moving mitochondria in the presence of Hep$^{CA}$ was reduced in aCC and RP2 and increased in VUMs (Fig. 4). We also noticed the presence of a subpopulation of fast-moving mitochondria that could reach speeds of up to 3 μm/s. Similar to the wild-type, we could not distinguish differences between anterograde and retrograde transport. In conclusion, vesicle transport to synaptic terminals does not seem to be affected by the upregulation of JNK activity, but mitochondrial mobility became abnormal, possibly due to altered energetic demands.

In order to determine the impact of JNK activity on motoneuron function, patch-clamp recordings were conducted on aCC and RP2 motoneurons, which receive similar cholinergic synaptic inputs[29].

Results show that expression of Hep$^{CA}$ in RN2+ neurons did not affect the amplitude or frequency of endogenous excitatory synaptic currents that these neurons receive [termed Spontaneously Rhythmic Currents (SRCs)] when voltage-clamped at −60mV (Fig. 5a and b). This suggests that the synaptic input to aCC/RP2 remained unaffected. However, current-clamp recordings, in which no holding potential was applied (I = 0), showed that the Resting Membrane Potential (RMP) of the aCC/RP2 motoneurons was significantly increased and their membranes hyperpolarized (Fig. 5c and d). Therefore, the average number of action potentials fired per SRC was notably reduced. This suggests that, while the synaptic drive was normal, the hyperpolarization induced by JNK resulted in reduced excitability and endogenous firing of these pioneer motoneurons.

### JNK signaling influences the axodendritic organization of *puc*-expressing neurons

The JNK signaling pathway does not seemingly affect synaptic vesicle transport or neuron receptive competencies but modulates their firing capability. It was possible that alterations to pioneer neuron morphology and dendrite integrity may contribute to these responses. To investigate this possibility, we evaluated the presence of structural defects of the *puc*-expressing pioneer neurons [RN2 (aCC, pCC and RP2)] in response to Hep$^{CA}$ overexpression.

aCC is a motoneuron whose axon gets incorporated into the intersegmental nerve (ISN)[30]. It displays a dendritic arbor of 8 to 10 dendrites[31] of an average length of 4 μm in a distinct region of the ISN myotopic dendritic domain. pCC is a medially-located interneuron with an anteriorly directed ipsilateral axon[32]. RP2 is a motoneuron, whose axon proceeds through the posterior root of the ISN[30,33] and branches out 7 to 9 dendrites of average length 4 μm (Fig. 6a).

The overexpression of Hep$^{CA}$ in RN2+ neurons caused frequent mislocalization of the soma of the aCC, pCC and RP2 cells towards the midline (compare Fig. 6a and b). The axons of these neurons also showed changes, including thickening (Fig. 5b), increased length (in pCC) (Fig. 6c), and altered morphology (Fig. 6d). JNK hyperactivation also impacted the density (Fig. 6e) and length (Fig. 6f and Supplementary Movie 5) of dendrites. Summarizing, increased JNK activity in specific pioneer neurons results in local structural and topographical disruptions, which could lead to incorrect 3D arrangements, cumulative misrouting, and potential impaired functionality.

### Different temporal roles for the JNK pathway

To determine if the roles of the JNK pathway modulating the structural organization and the functional optimization of the VNC, are causally related or independent, we evaluated the effects of altering JNK activity levels within the VNC at different times.

By using programmed temperature shifts between 29 and 18 °C or vice versa (see Methods), we selectively overexpressed Hep$^{CA}$ during early or late embryogenesis. Early (stages 10-13) hyperactivation of JNK in RN2+ neurons, before the initiation of neural activity[34], resulted in

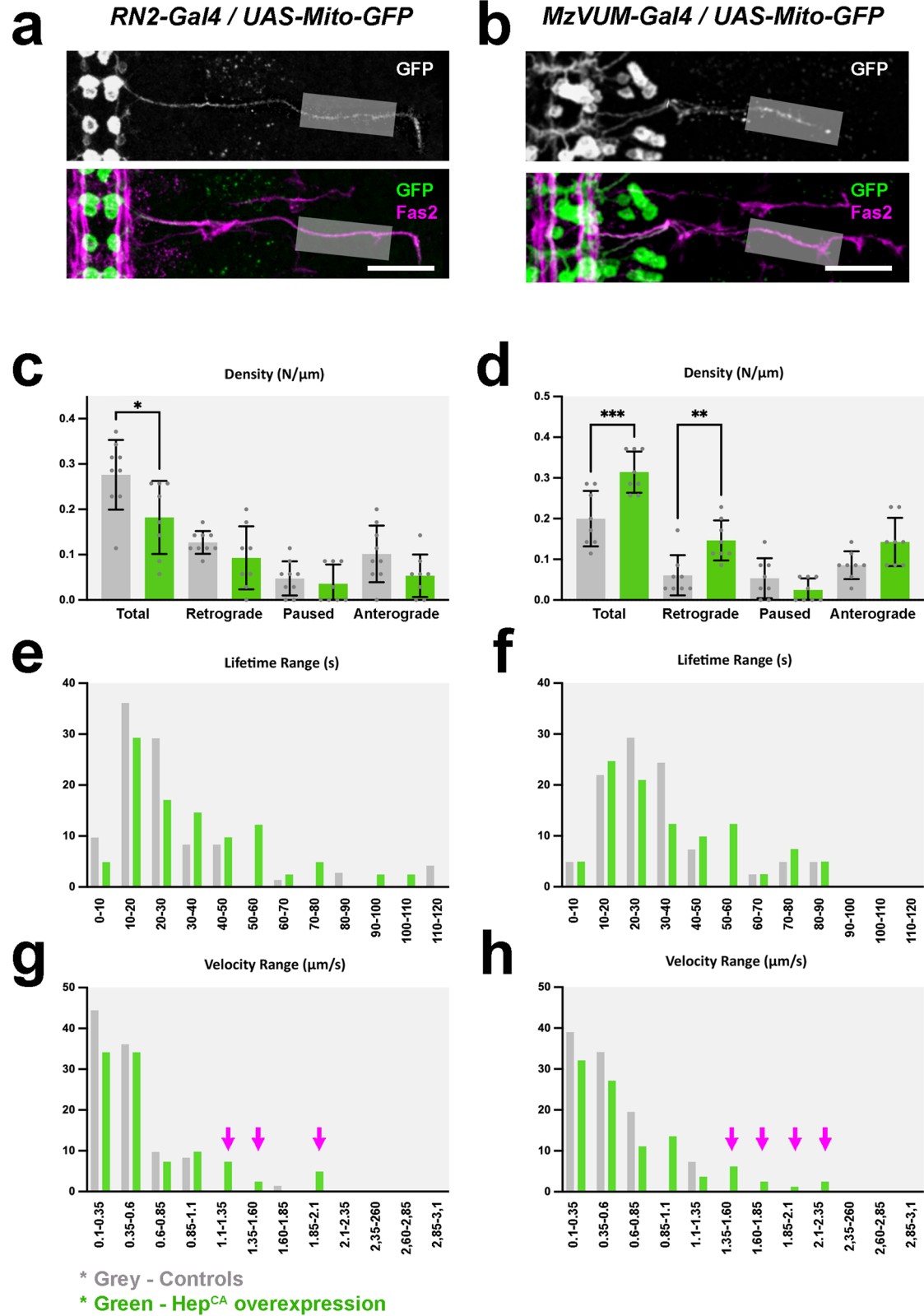

defects in the morphology of dendrites and axons, including axonal scaffold anomalies and the collapse of RN2+ neurons towards the midline (compare Fig. 7a, b and c). This also impacted VNC condensation (Fig. 7e). Further, although these embryos occasionally hatched, they displayed strong muscle coordination defects with an almost complete absence of bilateral peristalsis (Fig. 7f and Supplementary Movie 6). Conversely, late (stage 13-16) hyperactivation of the

JNK pathway resulted in the misalignment of RN2+ neurons which showed scarce and long dendrites, and formed heterogeneous axon bundles (Fig. 7d). This condition did not affect VNC condensation or motor coordination and peristalsis (Fig. 7e, g and Supplementary Movie 6).

In conclusion, axonal network organization, the condensation of the VNC, and the coordination of the peristaltic movements leading to

**Fig. 4 | JNK activity in early *puc* expressing neurons affects mitochondrial axonal transport. a** and **b** Flat-prepped, stage 16 embryos expressing Mito-GFP under the control of the RN2-Gal4 (**a**) and MzVum-Gal4 (**b**) lines immunostained for GFP (gray scale - top panels and green - bottom panels) and Fas2 (magenta - bottom panel). The grey rectangle marks the medial part of the ISN, where mitochondrial axonal transport was monitored (see Supplementary Movie 3). Scale bar is 25 μm. **c** and **d** Histograms showing mitochondrial density and directional motility in control (grey) versus Hep$^{CA}$ expressing (green) aCC/RP2 (**c**) and VUM motoneurons (**d**) ($n = 9$ for RN2 controls, $n = 8$ for RN2 experimental, $n = 9$ for MzVum controls and $n = 8$ for MzVum experimental). Data presented as mean ± SD. Parametric student *t*-tests were employed. Statistically significant differences were observed between control and JNK overactive neurons for both conditions [RN2 total (*$p = 0.0266$), RN2 retrograde ($p = 0.1884$), RN2 paused ($p = 0.5501$), RN2 anterograde ($p = 0.0966$), VUMs total (***$p < 0.0001$), VUMs retrograde (**$p = 0.0046$), VUMs paused ($p = 0.6968$) and VUMs anterograde ($p = 0.1003$). **e** and **f** Lifetime range distribution (expressed as per-unit) of motile mitochondria in aCC/RP2 (grey) and Hep$^{CA}$ expressing (green) embryos. **g** and **h** Mean velocity range distribution (expressed as per-unit) comparison of motile mitochondria in aCC/RP2 (**g**) and VUMs (**h**) motoneurons between control (grey) and JNK gain-of-function (green) conditions. For both sets of motoneurons the distribution was shifted to higher velocity classes in JNK hyperactive neurons.

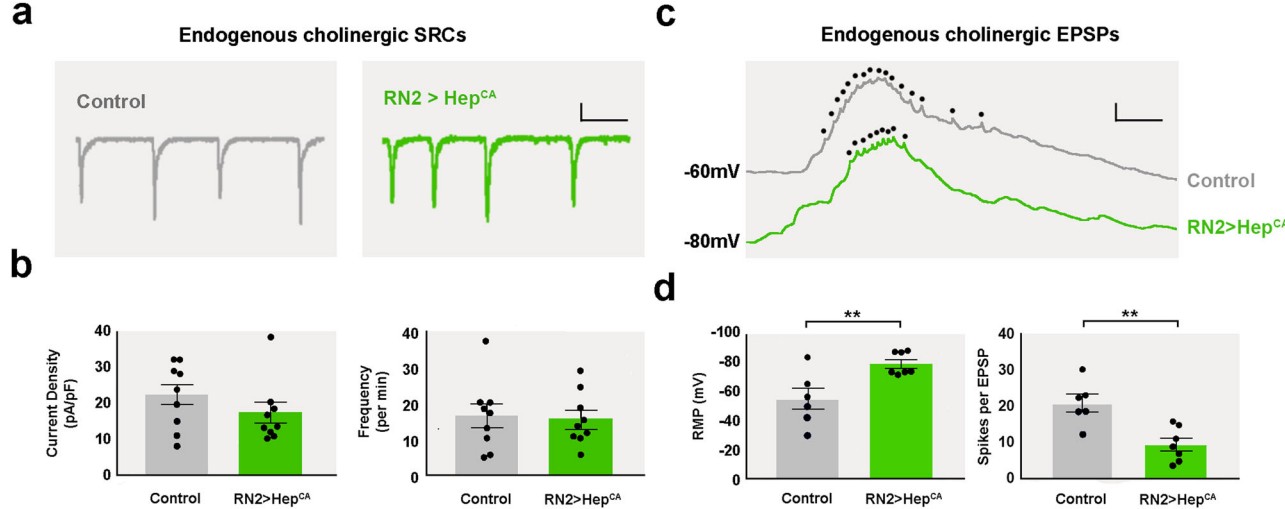

**Fig. 5 | JNK activity modulates neuronal firing. a** Representative traces of endogenously produced cholinergic SRCs from aCC/RP2 motoneurons in late stage 17 embryos in control (*RN2-Gal4 / CyO*) and experimental lines (*RN2-Gal4 > UAS-Hep$^{CA}$*). Scale Bar: 20 pA / 2 s. **b** No significant difference in SRC amplitude (21.7 ± 3.0 vs. 17.0 ± 3.1 pA / pF, control ($n = 9$) (gray) vs. *RN2-Gal4 > UAS-Hep$^{CA}$* ($n = 9$) (green) respectively, $p = 0.28$) or frequency (16.3 ± 3.3 vs. 15.4 ± 2.4 per min, $p = 0.83$) was observed. **c** Representative traces of endogenously produced cholinergic EPSPs from aCC/RP2 motoneurons in late-stage 17 embryos in control (*RN2-Gal4 / CyO*) and experimental lines (*RN2-Gal4 > UAS-Hep$^{CA}$*). Action potentials are indicated by dots. The amplitude of the action potentials is shunted because of the underlying depolarization. Scale Bar: 10 mV / 100 ms. **d** Motoneuron resting membrane potential (RMP) is significantly hyperpolarized in *RN2-Gal4 > UAS-Hep$^{CA}$* (−57.0 ± 7.6 vs. −81.5 ± 2.9 mV, control ($n = 6$) (gray) vs. *RN2-Gal4 > UAS-Hep$^{CA}$* ($n = 7$) (green) respectively, **$p = 0.009$). Consequently, the average number of action potentials fired per EPSP was also significantly reduced (20.6 ± 2.4 vs. 9.3 ± 1.7 spikes per EPSP, **$p = 0.003$). All data points are shown on graphs, bars represent mean ± sem.

hatching are dependent on precise levels of JNK activity in pioneer neurons at the onset of CNS morphogenesis. At these early stages, JNK modulates the establishment of the pioneer landscape and alterations in its activity induce aberrations in the axonal scaffold, cell bodies positioning, and muscle coordination. In late embryos, JNK affects the final refinement of neuron allocation and axonal tract formation, but is unrelated to motor control.

## Discussion
### JNK signaling coordinates motor activities
In the *Drosophila* embryo, muscles become active early and isolated twitches and unilateral waves of contraction occur before the generation of propagated action potentials[34]. This stochastic behavior evolves into organized coordinated muscle contractions that drive hatching. We asked if pioneer neurons and the architectural organization of the VNC have functional significance in this context. Our observations strongly suggest a close relationship between the morphogenesis of the VNC architecture and the coordination of motor function. Mutant embryos of different *puc* alleles and those overexpressing Hep$^{CA}$ in pioneer neurons exhibit severe structural defects[7] and slower, uncoordinated muscle twitching, resulting in the inability to hatch. This function is restricted to the *puc* + aCC, pCC, RP2 and VUM neurons and not observed when JNK signaling is altered in other neurons.

JNK signaling has previously been shown in *Drosophila* to promote dendrite pruning of sensory neurons[17]. Additionally, at the larval neuromuscular junctions (NMJ), JNK signaling affects the number of boutons and changes the amplitude of both excitatory junction potentials (EJP) and spontaneous mini EJPs. Sustained postembryonic JNK activation triggers synaptic growth but reduces the strength of synapses[9,15]. In fact, JNKs may play a role in developmental neuroplasticity, as they are activated by environmental stimulation and via induced electroconvulsive seizures in mice[21]. In the developing VNC, notable effects of increased JNK activity in pioneer motoneurons, to which the failure of motor coordination might be associated, are the aberrant dendritic arborizations and the hyperpolarization of their cell membranes (as shown in Fig. 5).

Dendrites carry the majority of synaptic inputs and their morphology and properties are critical for the propagation of postsynaptic potentials[35]. The synaptic integration in RN2 neurons may be affected by the changes in dendritic arborization and morphology caused by high JNK activity. Conceivably, this may lead to alterations on the frequency or capability to fire action potentials. In fact, it is well established that the membrane potential can control cell differentiation, cell shape and the arrangement of cortical tissue. Hyperpolarizing conditions can direct to a decrease in the number of connections in immature neurons in vitro[36]. In this scenario, the hyperpolarization of RN2 cells, following Hep$^{CA}$ overexpression, may lead to their abnormal

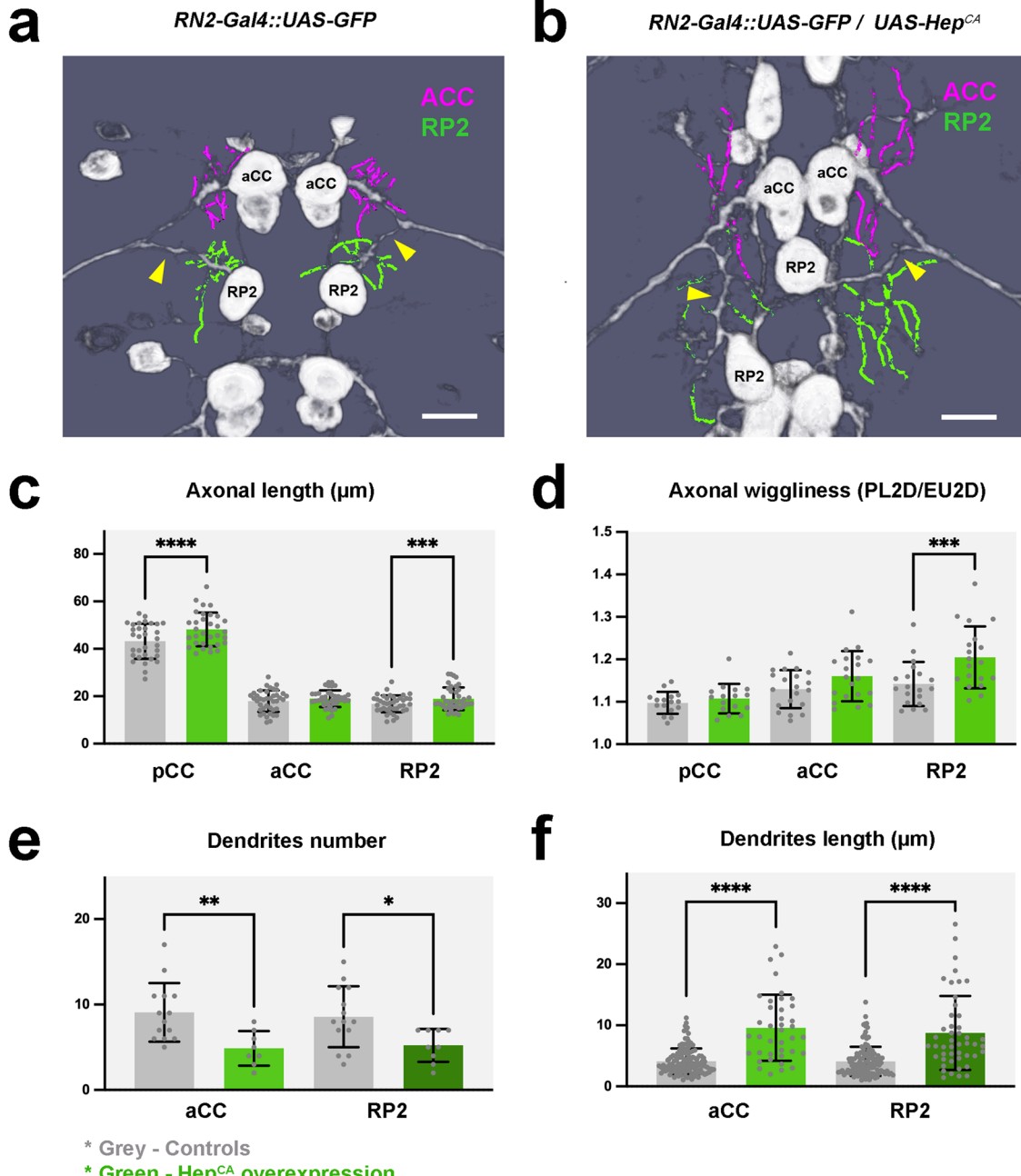

**Fig. 6 | The axonal and dendritic landscape of pioneer neurons is altered upon JNK hyperactivation. a** and **b** Flat-prepped, stage 16 embryos expressing mCD8-GFP (**a**) and mCD8-GFP and Hep[CA] (**b**) under the control of the RN2-Gal4 line immunostained for GFP. Traces of the aCC (magenta) and RP2 (green) proximal dendrites are over-imposed. Yellow arrowheads point to differences in RP2 axon morphology. Anterior is up. Scale bar is 10 μm. **c** Histograms showing the mean axonal length ±SD from *n* = 4 embryos of RN2 controls and *n* = 4 embryos of RN2/Hep[CA] for the pCC (*n* = 8 axons for controls and *n* = 8 axons for experimental), aCC (*n* = 10 axons for controls and *n* = 10 axons for experimental) and RP2 (*n* = 10 axons for controls and *n* = 9 axons for experimental) neurons. Statistically significant differences in the axonal length of the pCC and RP2 were detected between control and Hep[CA]expressing neurons. pCC length (****p < 0.0001), aCC length (p = 0.5032) and RP2 length (***p = 0.0006). Control embryos are represented in grey and Hep[CA]

expressing embryos in green. **d** Histograms showing the average axonal wiggliness ±SD from *n* = 4 embryos of RN2 controls and n = 4 of RN2/Hep[CA] embryos for the pCC (*n* = 8 axons for controls and n = 8 axons for experimental), aCC (*n* = 10 axons for controls and *n* = 10 axons for experimental) and RP2 (*n* = 8 axons for controls and n = 9 axons for experimental) neurons. RP2 axonal wiggliness was found significantly increased after Hep[CA] expression. pCC wiggliness (p = 0.3662), aCC wiggliness (p = 0.0785) and RP2 wiggliness (***p = 0.0041). **e** and **f** Graphs representing the average Dendrite Number (**e**) and average Dendrite Length (**f**) of the aCC and RP2 neurons ±SD. For both parameters significant differences were scored in the comparison of control (*n* = 14) and Hep[CA] (*n* = 8) expressing neurons. aCC dendrite number (**p = 0.0065), RP2 dendrite number (*p = 0.0266), aCC dendrite length (****p < 0.0001) and RP2 dendrite length (****p < 0.0001). Parametric student *t*-tests were used to detect statistically significant differences in all cases.

dendritic arborization. The direct impact of hyperpolarization on motor coordination, however, is less likely. So forth, the hyperpolarization of RN2 cells would have to be associated to early JNK functions and this is not the case, as the generation of action potentials in the embryo occurs at late embryonic stages[34].

TNT and Kir[2.1] overexpression differentially affect the functional properties of neurons. While TNT blocks synaptic vesicle release, Kir[2.1] causes membrane hyperpolarization. TNT in RN2 or CQ2 cells does not modify the overall organization of the VNC axonal network or the motor coordination of the embryo, albeit

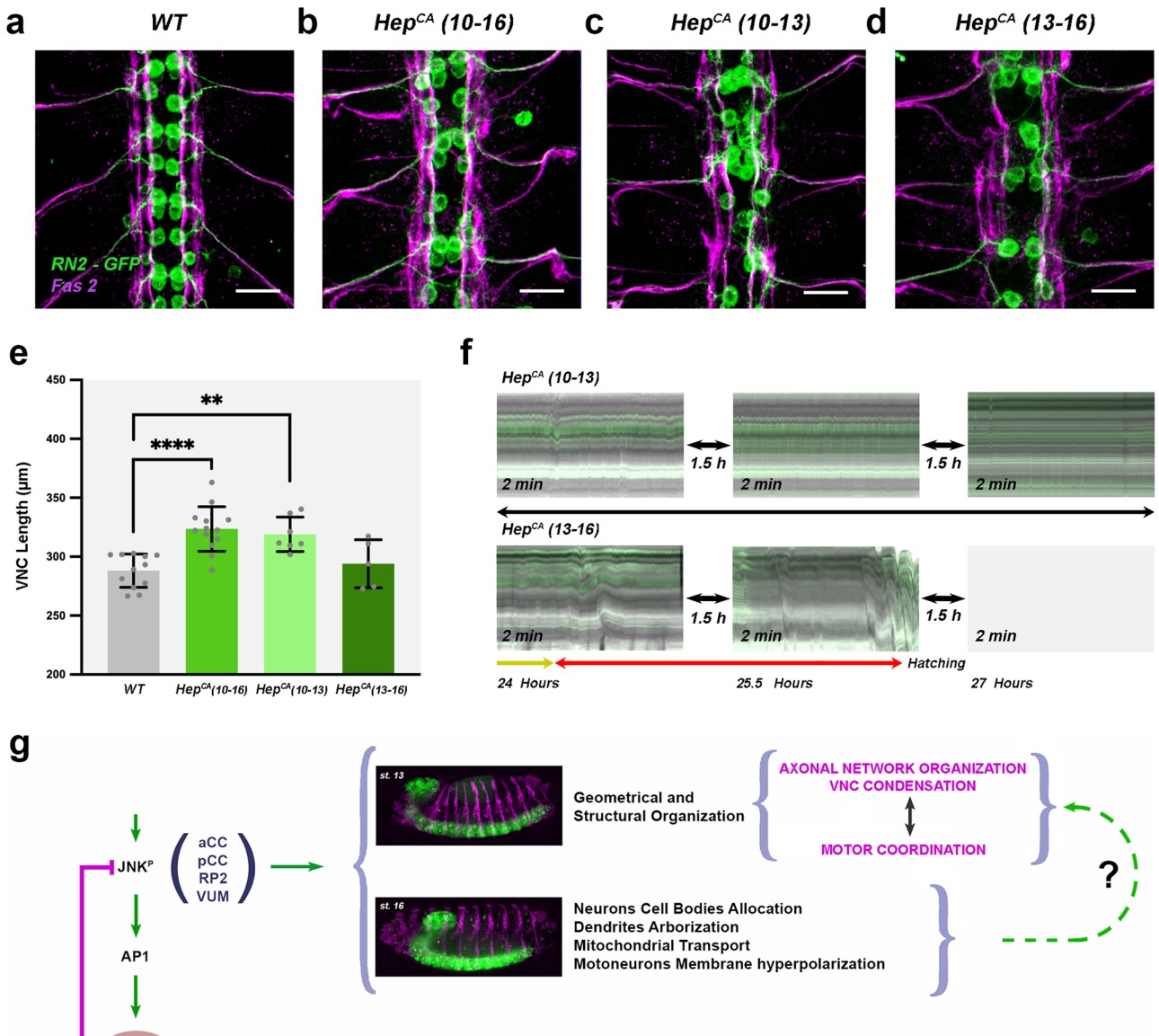

**Fig. 7 | JNK activity modulates VNC condensation and embryo motor coordination. a–d** Fas 2 (magenta) and GFP (green) immunoreactivity of late stage 16 RN2-Gal4 embryos subjected to different temperature shift conditions and expressing: (**a**) GFP (WT) (n = 12) (maintained at 29 °C continuously from stage 10 to 16); (**b**) GFP together with a *Hep*$^{CA}$ transgene; Hep$^{CA}$ (10-16) (n = 13) (maintained at 29 °C continuously from stage 10 to 16); (**c**) Hep$^{CA}$ (10-13) (n = 7) (kept at 29 °C between stages 10 and 13) and (**d**) Hep$^{CA}$ (13–16) (n = 6) (kept at 29 °C during stages 13 and 16). Maximum projection of ventral views across three VNC abdominal segments. Scale Bar is 10 μm. Overexpressing *Hep*$^{CA}$ at early stages affects pioneer functions and provokes the collapse of neuronal cell bodies into the midline. At late stages, dendrite arborizations, axons fasciculation and cell bodies alignment are disturbed. **e** Quantification of the VNC length in μm (average ±SD) for each condition above. Statistically significant differences in length were detected between WT (n = 12) and Hep$^{CA}$ (10–16) (n = 13) (****p < 0.0001) and Hep$^{CA}$ (10-13) (n = 7) embryos (**p = 0.0016) but not with Hep$^{CA}$ (13–16) (n = 5) (p = 0.8661).

**f** Representative kymographs displaying muscle profiles between 21 and 25 hours AEL for the left (L) side of embryos expressing Hep$^{CA}$ in RN2+ cells at early (10-13) or late (13-16) developmental stages. Embryos overexpressing *Hep*$^{CA}$ at early stages in RN2+ neurons (n = 12) never managed to coordinate muscle movements but those overexpressing *Hep*$^{CA}$ late (n = 8), developed full peristalsis (red arrows) and hatched in all occasions (Supplementary Movie 6). **g** Precise JNK activity levels in early-specified neurons (at least aCC, pCC, RP2 and VUMs) are regulated by a negative feedback loop mediated by Puc. Excessive JNK activity in *puc* mutants or upon overexpressing *Hep*$^{CA}$ in the same neurons leads to autonomous defects in axonal paths, dendrites' number and shape, mitochondrial axonal transport and hyperpolarization of motoneurons' membranes. Further, the architectural robustness of the VNC is affected, its condensation prevented and the coordination of embryonic motor activities and peristaltic movements inhibited. The aberrant axonal network organization and the incomplete VNC condensation may affect wiring optimization and eventually elicit motor uncoordination.

impeding hatching. By contrast, Kir$_{2.1}$ expression does alter the 3D architecture of the VNC axonal network, but, opposite to JNK and without compromising shortening, increases its robustness (Fig. 3). Actually, it has been previously found that Kir$_{2.1}$ activity can affect cell specification and lead to morphogenetic defects[37–40]. In the wildtype, both, RN2 and CQ2 neurons' dendritic arborizations extend into medial domains in the neuropile[41]. We observed for

RN2 that this arborization is affected after Kir$_{2.1}$ overexpression (as it is CQ2 cells survival), suggesting that these structural defects might be causal of these embryos VNC hypercorrelative axonal network organization. While motoneuron hyperpolarization seems to be a common consequence of the overexpression of Kir$_{2.1}$ and Hep$^{CA}$ in RN2 cells, they differ in the way they affect the structural organization of the VNC and motor outcomes.

RN2 cells overexpressing Kir$_{2.1}$ do not show VNC condensation or motor coordination defects disentangling these processes from membrane hyperpolarization. Our results thus suggest that muscle coordination failure is primarily associated to the specific structural defects stemming from the early hyperactivation of the JNK pathway in the pioneer neurons of the VNC (Fig. 7g).

### JNK and the cell biology and physiology of pioneer neurons

The structural organization of the VNC appears to have an early direct role on embryos' motor coordination. This structural organization is affected by the level of JNK activity on pioneer neurons and the eventual suppression of Fas2 expression[7]. Indeed, Fas2 expression suppression does prevent the progression of motor patterns' maturation (Supplementary Fig. 4). It is known that the embryonic development of cholinergic central synapses[42], like the formation of the glutamatergic Neuromuscular Junctions (NMJ)[26], does not require Fas2, ruling out the possibility that defects in synapse formation in Fas2 mutant conditions could have a functional involvement in motor coordination progression. Yet, the normal pattern of synaptic connectivity is, nevertheless, sensitive to changes in the relative levels of expression of Fas2 in pre and postsynaptic neurons[42] or muscles[43]. Thus, it would still be possible that any experimentally elicited Fas2 expression imbalance may cause alterations in synaptic inputs to motor neurons or to the muscles concerned. The fact that in the Fas2 (and JNK) mutant combinations employed the reduction of Fas2 levels appeared to be constitutive makes this last option unlikely.

Furthermore, the JNK cascade in the CNS regulates microtubule dynamics by phosphorylating both microtubule-associated proteins, including Tau[44], or the Kinesin Heavy Chain, which induce the kinesin motor to release from microtubules[13]. In the *Drosophila* VNC, Tau, together with microtubule-binding spectraplakins, modulates microtubule stability which leads to alterations of JNK signaling activity, consequently affecting kinesin-mediated axonal vesicle transport[10]. In our hands, however, interfering in RN2 pioneer motoneurons, JNK activity does not significantly affect synaptic vesicles transport (Supplementary Fig. 4). Yet, it inhibits mitochondrial motility (Fig. 4), an effect previously observed in third instar larvae, where the JNK pathway mediates the inhibition of mitochondrial anterograde transport by oxidative stress[14]. This suggests that the influence of JNK signaling in orchestrating vesicle movements is probably cell and developmental stage specific. Significantly, it indicates that synaptic vesicle axonal transport, in pioneer neurons, is irrelevant for embryonic motor coordination.

While altering JNK signaling affects the cell biology (axonal transport) and physiology (hyperpolarization) of pioneer neurons, we do not know if it does so through its well-established role regulating gene expression, phosphorylating substrates such as Jun or histone modifiers, or, alternatively, acts via the phosphorylation of cytoplasmic targets. The fact that, in different models, the JNK pathway mediates microtubule destabilization leading to a transport roadblock by phosphorylating cytoplasmic substrates, including microtubule-associate proteins[44,45], strongly supports this last possibility.

### Wiring optimization

It is established that spontaneous activities, that persistently stimulate postsynaptic cells, result in long-term synapse potentiation[46]. These activities propagate efficiently between nearby cells, ensuring connections are strengthened, whereas those from distant cells are lost[47]. Bringing neurons in proximity improves functional interconnectedness and improves wiring optimization[2]. We suggest that these classic principles are followed during the coordination of motor activities in the *Drosophila* embryo. Several arguments support this premise. First, both axonal network organization and motor coordination require a precise level of JNK activity at early developmental stages, enabling activity of pioneer neurons (Fig. 7). Second, either activation or

ablation of the EL interneurons disrupt bilateral muscle contractions in the larvae[48]. These ELs make presynaptic contacts with the RP2 motoneurons, which we found critical for both architectural organization of the VNC[7] and for motor coordination (Fig. 2 and 7g). Third, ELs, and the dorsal projecting motoneurons they innervate, are nearly all *puc*-expressing neurons[7].

We foresee that the principles of wiring optimization may also apply at a postembryonic level. As larvae develop, their body area grows by over two orders of magnitude[49]. To accommodate such a change, mechanosensory neurons must extend their dendrites and increase their receptive field[50], while motoneurons add more synapses to adapt for larger muscles[49,51]. During scaling, synaptic contacts between nearby neurons are sustained[52]. In this scenario, a postsynaptic neuron can only connect with nearby, precisely positioned, pre-synaptic sites. The propensity to form stable local functional synapses appears to operate across development.

Here we have uncovered an intimate causal link between the temporal control of the activation of the JNK pathway in pioneer neurons, the correct structural organization of axonal networks, the eventual condensation of the VNC, and embryonic motor coordination (Fig. 7g). Correlative geometrical and mechanical studies and further cell type-specific genetic interference analyses will be necessary to dissect the precise mechanisms underlying this process. Yet, wiring optimization across different circuits, by enhancing partners' proximity and topographical robustness, appears to be an essential character associated with the morphogenesis and functionality of the nervous system.

## Methods

### Drosophila strains
The following stocks were used:

 w[1118]; puc$^{E69}$LacZ/TM3, twi-GFP[53]

 w[1118]; P{w[+mC]=UAS-Hep.Act}2 (BDSC #9306)

 w[1118]; P{w[+mC]=eve-Gal4.RN2}T, P{w[+mC]=UAS-mCD8::GFP.L}; P{w[+mC]=eve-Gal4.RN2}G, P{w[+mC]=UAS-mCD8::GFP.L} (Dr. Irene Miguel-Aliaga)

 w[1118] P{w[+mW.hs]=GawB}MzVUM; P{y[+t7.7]w[+mC] = 10xUAS-IVS-mCD8::GFP} attP40 (Dr. Irene Miguel-Aliaga)

 w[1118]; G2O3:UAS-TNT; ZCL2144[8] (Dr. Matthias Landgraf)

 w[1118];; ZCL2144[8] (Dr. Matthias Landgraf)

 w[1118]; P{UAS-syt.eGFP}2 (BDSC #6925)

 w[1118]; P{w[+mC]=UAS-mitoGFP.AP}2 / CyO (BDSC #8442)

 w[1118]; P{w[+mC]=UAS-Hsap\KCNJ2.EGFP}7 (BDSC #6595)

 w[1118] P{w[+mC]=UAS-bsk.DN}2 (BDSC #6409)

In all cases, unless otherwise stated, embryos of the *w1118* strain served as controls.

### Genetics

All crosses were performed at room temperature and after 48 hours were shifted to different temperatures as the individual experiments required.

For temperature shifts, 2 hours embryo collections were done at room temperature in agar plates and aged for a further 2 hours. Then, they were transferred, either to a 29 °C incubator for 4 hours and then to an 18 °C incubator for 12 additional hours [Hep$^{CA}$ (10-13)] or to an 18 °C incubator for 12 hours and then to a 29 °C incubator for 4 hours [Hep$^{CA}$ (13-16)]. These timing regimes were calculated to enhance Hep$^{CA}$ expression during the early pioneering stages or during the final cell allocation and axonal refinement stages of VNC development. After the temperature shifts the embryos were processed for immunocytochemistry or for live imaging.

### Immunohistochemistry

Immunostaining of flat-prepped stage 16 *Drosophila* embryos was performed using the primary antibodies: mouse anti-Fas2 (1:100, DSHB, #1D4 anti-Fasciclin II, RRID:AB_528235), and rabbit anti-GFP tag

polyclonal (1:600, Thermo Fisher Scientific, # A-11122). Immunostaining of whole mount stage 17 *Drosophila* embryos was done using a rabbit anti-*Drosophila* muscle myosin sera[54] at 1:500 dilution (a gift from Dan Kiehart).

The antibodies used for detection were: Goat anti-Rabbit IgG (H + L), Alexa Fluor 488 conjugate (Thermo Fisher Scientific, #A-11008) and Goat anti-Mouse IgG (H + L), Alexa Fluor 555 conjugate (Thermo Fisher Scientific, #A-21422). All secondary antibodies were used in a dilution of 1:600.

### Sample preparations for immunodetection and image acquisition

*Drosophila* embryo dissections for generating flat preparations were performed according to[30]. Briefly, flies maintained in apple juice-agar plates at 25 °C were synchronized by repetitive changes of the juice-agar plate, with a time interval of 2 hours. All embryos laid within this time window were aged at different temperatures following experimental requirements until reaching mid-stage 16 (3-part gut stage). At this point embryos were dechorionated with bleach for 1 min, poured into a mesh and rinsed extensively with water. For dissection, embryos were transferred with forceps on the surface of a small piece of double-sided tape, adhered on one of the sides of a poly-L-Lysine coated coverslip. After orienting the embryos dorsal side up and posterior end towards the center of the coverslip, the coverslip was flooded with saline (0.075 M Phosphate Buffer, pH 7.2). Using a pulled glass needle the embryos were manually de-vitelinized and dragged to the center of the coverslip, where they were attached to the coated glass with their ventral side down. An incision on the dorsal side of the embryo was performed using the glass needle from the anterior to the posterior end of the embryo. The gut was removed by mouth suction and a blowing stream of saline was used to flatten their lateral epidermis. Tissue fixation was done with 3.7 % formaldehyde in saline for 10 minutes at room temperature. After this point standard immunostaining procedures were followed.

Image acquisition was performed on a Zeiss LSM 700 inverted confocal microscope, using a 40X oil objective lens (NA 1.3). Z-stacks, spanning the whole VNC thickness, were acquired sectioning with a step size of 1 μm. Dendrites visualization was achieved with an Andor Dragonfly 505, using a 100X oil objective lens (NA 1.3) and parallel deconvolution. Image processing was performed with Fiji[55] and Imaris Cell Imaging Software (Oxford Instruments).

### Live imaging

Dechorionated stage 17 embryos were glued ventral side down on a MatTek glass bottom dish and they were covered with S100 Halocarbon Oil (Merck) to avoid desiccation. Image acquisition was performed on a Zeiss LSM 780 inverted confocal microscope, using a 25X oil immersion lens (N.A 0.8, Imm Korr DIC M27). Processing of time-lapse data was done with Fiji[55].

### Image analysis

Image analysis and quantification of fluorescence intensity were performed using Fiji[55]. In immunostainings where quantification of the fluorescent intensity was required, extra care was taken so that the same antibody batch was used for staining embryos of different genotypes, while identical confocal acquisition settings were applied.

### Correlations data analysis

3D correlation analyses were performed as in[7]. Images were scaled to be isotropic in all axes. Viewed along the AP axis, images were cropped to include only the VNC, and then the VNC was split into 50 bins, corresponding to 1.65 μm length each. Within each bin, a maximum intensity projection was performed along the included planes in the z-axis. The Matlab function *imregister* was then used to perform image registration.

### Statistical analysis

All statistical analyses were performed using GraphPad Software (GraphPad Software Inc., La Jolla, CA, USA). In all cases one sample t test or one way ANOVA for multiple comparisons were performed and probability values $p < 0.05$ were considered as significant.

### Muscle contraction analysis

Stage 17 embryos carrying the G203, ZCL2144 or both GFP muscle markers were dechorionated and mounted in halocarbon oil (3:1 mix of S700 and S27) following standard procedures. Muscle contractions were monitored at the embryo's middle plane, using a 25X oil immersion lens (N.A 0.8, Imm Korr DIC M27) in a Zeiss LSM780 confocal microscope. For each acquisition 1000 frames were obtained with a scanning speed of 200 msec per frame with a time interval between acquisition cycles of 15 minutes. Muscle activity was monitored up to embryo's hatching or for a total of 12 hours at room temperature. For the experiments where Hep$^{CA}$ over-expression in RN2 and MzVum neurons was required, both control and Hep$^{CA}$ overexpressing embryos were filmed at 29°C for equal time periods. For temperature-shift analyses of animals expressing Hep$^{CA}$ in RN2 neurons, embryos of equivalent pre-hatching age were recorded at 200 msec per frame for 500 frames. Post-imaging, movies were processed with Fiji[55] and Kymographs were generated using the Fiji plugin KymoResliceWide.

### Axonal trafficking

To examine axonal trafficking, stage 17 embryos expressing UAS-Mito-GFP or UAS-Syt-GFP were dechorionated and mounted laterally in halocarbon oil (3:1 mix of S700 and S27) for live imaging. Mytochondria and synaptic vesicles transport were monitored in the medial part of the Inter-Segmental Nerve (ISN) using a 40X oil immersion lens (N.A 1.3, Oil DIC M27) in a Zeiss LSM780 confocal microscope. A constant frame size of 512 (35 μm) x100 pixels, with an XY zoom of 6 and a recording line spacing of 2 were routinely employed. Single plane movies were acquired from ISN aligned to the longest axis of the frame with a scanning speed of 1.11 sec, with 1 sec interval for a total of 90 cycles. All movies were processed with Fiji and Kymographs were generated using the Fiji plugin KymoResliceWide.

The Mean Velocity and the Lifetime parameters for each particle were calculated from the Kymographs as follows. The particles trajectories were traced and saved as independent ROIs. The xy coordinates for each ROI were then extracted using the getSelectionCoordinates function in Fiji of the custom made Macro coordinates.ijm shown below:

```
run ("Fit Spline", "straighten");
getSelectionCoordinates (x, y);
x2 = 0; y2 = 0; distance=0;
for (i = 0; i < x.length; i ++)
if (i > 0) {dx = x[i] - x[i−1]; dy = y[i] - y[i−1];
distance = sqrt(dx*dx+dy*dy);}
print (i, x[i], y[i], distance);}
```

These *xy* coordinates represent the particles displacement in both space (*x* values) and time (*y* values), thus permitting the calculation of Mean Velocity and Lifetime.

### Analysis of axonal and dendrite morphology by segmentation

To describe the axonal and dendrite morphology of the RN2+ neurons, we analyzed images of flat-prepped embryos immunostained against GFP and Fasciclin 2 using the Simple Neurite Tracer Fiji plugin[56]. Image segmentation by this tool allows a semi-automated tracing of axons and dendrites in 3D, providing measurements of total axonal and dendrite length, as well as of dendrite number directly from skeletonized images.

### Embryo whole-cell patch-clamp recordings

Dissections and recordings were performed as described[34] at room temperature (20-22 °C). Late stage 17 *Drosophila* embryos were

dissected in external saline (in mM: 135 NaCl, 5 KCl, 4 MgCl$_2$·6H$_2$O, 2 CaCl$_2$·2H$_2$O, 5 N-Tris[hydroxymethyl]methyl-2-aminoethanesulfonic acid, and 36 sucrose, pH 7.15). The CNS was exposed through a dorsal-longitudinal cut of the dechorionated embryonic body wall musculature. Embryonic cuticle was secured to a Sylgard (Dow-Corning, Midland, Michigan, USA)-coated cover slip using tissue glue (GLUture; WPI, Hitchin, UK) and larval gut and fat body removed. The glia surrounding the CNS was partially removed using protease (1% type XIV; Sigma, Dorset, UK) contained in a wide-bore (15 μm) patch pipette. Whole cell recordings were carried out using borosilicate glass electrodes (GC100TF-10; Harvard Apparatus, Edenbridge, UK), fire-polished to resistances of between 16-20 MΩ. Cell identity was confirmed through RN2-GAL4 driven expression of UAS-CD8-GFP. Recordings were made using a MultiClamp 700B amplifier. Endogenous excitatory synaptic cholinergic input to the aCC/RP2 motoneurons were recorded in voltage clamp and current clamp sampled at 20 kHz and low pass filtered at 10 kHz, using pClamp 10.6 (Molecular Devices, Sunnyvale, CA). Only neurons with an input resistance of ≥ 1GΩ were accepted for analysis.

### Reporting summary

Further information on research design is available in the Nature Portfolio Reporting Summary linked to this article.

## Data availability

The authors declare that all data supporting the findings of this study are available within the Article, its Supplementary Information and Source Data files, and are also available from the corresponding authors upon request. Source data are provided with this paper.

## Code availability

The correlation analysis was performed in Matlab using standard Matlab functions. Full code is provided on Github [https://github.com/TimSaundersLab][7].

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

## Acknowledgements

We are extremely grateful to all colleagues that have provided us materials, training and guidance throughout this work as well as for commenting on the manuscript. Amongst them we like to specially thank Christian Klambt, Claude Desplan, Matthias Landgraf, Chris Doe and Brian McCabe. We also thank our colleagues at the IBMB, especially Elena Rebollo at the Molecular Imaging Platform (MIP), and the BSRC "Alexander Fleming" for their encouragement and support. The Martín-Blanco laboratory was supported by funds from Ministry of Innovation and Science (BFU2014-57019-P, BFU2017-82876-P and PID2020-116273GB-I00) and Fundación Ramón Areces. R.A.B. was supported by a BBSRC BB/L027690/1 grant and S.W.V. by a BBSRC CASE studentship. Work on this project benefited from the Manchester Fly Facility with funds from University and the Wellcome Trust (087742/Z/08/Z).

## Author contributions

K.K., R.A.B., G.P. and E.M-B. designed the experiments and analyzed the data. K.K. and E.M-B. conducted the experiments. S.W.V. and R.A.B. performed the electrophysiological analysis. K.K. and E.M-B. wrote the initial draft, R.A.B. and G.P. revised the final draft.

## Competing interests

The authors declare no competing interests.
