## [Peer Review File · Nature Communications]

REVIEWER COMMENTS

Reviewer #1 (Remarks to the Author):

Key results

The paper titled “Puckered in pioneer neurons coordinates the motor activity of the *Drosophila* embryo” provides novel insights into the role of June Kinase (JNK) signaling pathway in pioneer neurons during embryonic development. Puckered (Puc), a phosphatase, plays an important role in regulating the June Kinase (JNK) pathway and its activity in puc-expressing neurons via a negative feedback loop. The findings of the study elucidate that the loss of puc in puc+ neurons leads to an increase in JNK signaling. At early embryonic stages, the elevated JNK signaling disrupts vnc condensation and motor coordination, while at late embryonic stages, overactivation of JNK leads to dendrite and axon morphology defects, cell body mis-localization, membrane hyperpolarization and anomalies in mitochondrial transport. These effects are then confirmed to be directly from increased JNK signaling because functional perturbation (using TNT or Kir2.1) of pioneer neurons did not affect vnc condensation or motor coordination. It is then implicated that most of these phenotypes could contribute to JNK’s role in wiring optimization.

Significance

Building upon their recent Nature Comms paper (Karkali et al 2023), in this manuscript, the authors describe more phenotypic changes in embryonic motor neurons caused by JNK signaling. As a descriptive paper, it has advancements in the types of cellular changes to build on previous observations of changes in VNC architecture. There was some discussion of potential cellular mechanisms of how these phenotypes act together and possibly influence each other, but nothing that can be proven with the data shown. Additionally, there is no mention of any molecular mechanisms, which were explored in their recent paper.

Validity

I didn’t notice any glaring issues with the methodology, data, or interpretation that would invalidate the results or conclusions.

Suggestions and Clarification Points

- The authors have shown that altering JNK activity in pioneer neurons affects both VNC condensation and embryonic motor coordination. It would be interesting to investigate whether JNK

activity also plays a role in other aspects of CNS development such as synapse formation and pruning. This could be done by examining the morphology and connectivity of *puc+* neurons with altered JNK activity during later stages of embryonic development.

- The authors could also explore the molecular mechanisms underlying the abnormal structural organization in the VNC and determine whether it is a direct consequence of altered JNK signaling or an indirect effect of impaired motor coordination.
- Add Fas 2 VNC images next to correlation graphs (like in the previous paper) to make the interpretation more accessible to those who haven't seen something like this before
- In the discussion the authors are hypothesizing how the late-stage morphological changes and resting membrane potential may be interlinked and then go on to make the point that although this changes the property of the neuron, because these changes are not seen till the later stages they cannot be the cause of the lack of motor coordination in early stages. However, the way it was stated is convoluted and it took me a few rereads to understand what they were saying.
- In the results on changes in motor coordination, the authors state that the phenotype is not due to loss of muscle integrity. There is no explanation as to how they came to that conclusion, so a statement on why they believe that would help.
- The authors claim that some phenotypes only show up in later stages, but only show the later time point where we can observe the phenotype. There is also no mention of testing earlier time points that did not show the phenotype.
- Loss of JNK signaling also showed failure in VNC condensation in their previous paper, but only gain of signaling was used in this paper, whether that be disinhibition through puckered mutant or activation with HepCA. Given that both scenarios give the same loss in VNC condensation, they could include results for the loss of signaling as well since any difference in phenotypes may help inform the exploration of the mechanism in the future.

Reviewer #2 (Remarks to the Author):

The relationship between the architecture and functionality of the nervous system remains a topic of intense interest. Here, the authors investigate the relationship between JNK signaling and motor neuron development in the fruit fly embryo. The authors report that JNK signaling is required for morphogenesis of the ventral nerve chord (VNC), in part by the action it exerts on a negative feedback loop mediated by the dual specificity phosphatase Puckere.

Specifically, the authors investigated how JNK signaling autonomously regulates neuronal electrophysiological properties. They showed that constitutively overexpressing JNK in motor neurons affected their ability to coordinate the development of motor circuitry required for hatching.

Although the authors analyze motor neuron activity, embryo movement, and mitochondrial transportation in motor neurons in which they specifically overexpress HepCA, it is unclear whether the functional deficit described above is the result of architectural changes or alteration of motor neuron function. In other words, the authors have not ruled out the possibility that the functional defects found in HepCA overexpressed motor neurons are due to structural defects in the mutants specifically, rather than to JNK signalling.

Some of the data in the manuscript is unconvincing, and most of the experiments are not well quantified.

Specific Points:

1. In general, most of the experiments are not well quantified. Although the authors show some statistics, most of the time they do not clearly indicate the number of animals or sample sizes. For example, in Figure 1, the authors describe the contraction phenotypes by Stages A to E in the main text and show the evolution of the contraction waves graphically in Supplemental Figure 1. However, the authors provide no details on how the features they propose contribute to the different stages of contraction, for example, by using the features for segmentation of Stages A to E. Similarly, although the mutant phenotype is clear, the authors present little if any quantitative data. For example, they could provide the average duration of each stage, as well as the number of samples tested. Without sample sizes, it is difficult for any reader to assess the statistical significance of the findings. (For example, how many samples were tested in Figure 1C and D, Figures 2, 3, 4, etc.?)

2. Most of the experiments involve overexpression of HepCA, Does RNAi knockdown of puc in motor neurons RN2 or CQ21 similarly show HepCA overexpression, electrophysiological phenotypes, and/or mitochondrial transportation in motor neurons?

3. When Kir2.1 is expressed in motor neurons RN2 or CQ2, is the morphology of motor neurons affected (Supplemental Figure 3). Please quantify how often each phenotype occurred.

4. Figure 4: Please indicate how many animals were used, as well as the number of mitochondria-positive dots. Also, please indicate the sample size. (For example, is $N = 1$? If so, is the average and bar necessary?). Furthermore, please indicate the type of statistical tests that were used to obtain p-values that show differences between genotypes (for example, in panels c to h).

5. I am wondering what is the mitochondrial transport phenotype upon RNAi knockdown of Jnk signaling in RN2?

Figure 6 c - f: It is difficult to see how p-value of < 0.001 can be obtained with a sample size of 4 with the substantial overlap of standard deviations between genotypes. Please confirm the accuracy of this statistical analysis.

Minor points

1. Line109: Please explain what U1 – U5 motor neurons refer to.

2. The figures could be improved by adding some legends. For example, in Figure 4a – h, it would be helpful to provide legends for the color code and genotype (for example, gray bar: control; green bar: HepCA > RN2; etc.).

Puckered in pioneer neurons coordinates the motor activity of the *Drosophila* embryo

Katerina Karkali Samuel W. Vernon, Richard A. Baines, George Panayotou and Enrique Martín-Blanco

REPLY TO REVIEWERS

Reviewer #1

Key results

The paper titled “Puckered in pioneer neurons coordinates the motor activity of the *Drosophila* embryo” provides novel insights into the role of June Kinase (JNK) signaling pathway in pioneer neurons during embryonic development. Puckered (Puc), a phosphatase, plays an important role in regulating the June Kinase (JNK) pathway and its activity in puc-expressing neurons via a negative feedback loop. The findings of the study elucidate that the loss of puc in puc⁺ neurons leads to an increase in JNK signaling. At early embryonic stages, the elevated JNK signaling disrupts vnc condensation and motor coordination, while at late embryonic stages, overactivation of JNK leads to dendrite and axon morphology defects, cell body mis-localization, membrane hyperpolarization and anomalies in mitochondrial transport. These effects are then confirmed to be directly from increased JNK signaling because functional perturbation (using TNT or Kir2.1) of pioneer neurons did not affect vnc condensation or motor coordination. It is then implicated that most of these phenotypes could contribute to JNK’s role in wiring optimization.

Significance

Building upon their recent Nature Comms paper (Karkali et al 2023), in this manuscript, the authors describe more phenotypic changes in embryonic motor neurons caused by JNK signaling. As a descriptive paper, it has advancements in the types of cellular changes to build on previous observations of changes in VNC architecture. There was some discussion of

potential cellular mechanisms of how these phenotypes act together and possibly influence each other, but nothing that can be proven with the data shown. Additionally, there is no mention of any molecular mechanisms, which were explored in their recent paper.

We disagree on qualifying this manuscript as descriptive. We provide evidence supporting a direct link between the VNC condensation/organization defects induced by JNK hyperactivation in pioneer neurons and the coordination of motor activities required for embryo hatching. The disorganization of the neuronal landscape and the failure of tissue packaging impose structural restrictions to connectivity and, as a consequence, result in muscle activity coordination defects. No signaling or flux of information seem to be involved and geometrical and structural constraints appear to be sufficient to disrupt normal behavior. The molecular mechanisms involved in altering the structural organization of the VNC were explored in our previous Nat Comm report.

Validity

I didn't notice any glaring issues with the methodology, data, or interpretation that would invalidate the results or conclusions.

We thank the reviewer for approving of our methodology, analyses and interpretations

Suggestions and Clarification Points

- The authors have shown that altering JNK activity in pioneer neurons affects both VNC condensation and embryonic motor coordination. It would be interesting to investigate whether JNK activity also plays a role in other aspects of CNS development such as synapse formation and pruning. This could be done by examining the morphology and connectivity of *puc+* neurons with altered JNK activity during later stages of embryonic development.

Axonal morphology has been studied; and dendritic number and density have been quantified. Further, we have also evaluated vesicle and mitochondrial axonal transport and the electrophysiological responses of the aCC/RP2 pioneer motoneurons. While connectivity and synapse formation and pruning could also be analyzed, this would demand to build a connectome by EM for the relevant cells in mutant conditions. The study of the EM volume is expected to yield important information and we have already set up a project with the AIC facility of Janelia Howard Hughes Institute (already funded) aiming to generate this data by FIB-SEM in experiments scheduled for April 2024. Yet, we consider that these analyses go far beyond the information needed to substantiate our conclusions.

- The authors could also explore the molecular mechanisms underlying the abnormal structural organization in the VNC and determine whether it is a direct consequence of altered JNK signaling or an indirect effect of impaired motor coordination.

When motor coordination is impaired by interference in neurotransmitter exocytosis (TNT overexpression) or by neurons electrophysiological polarization (Kir2.1 overexpression), the VNC condensation is largely unaffected (Figure 3). In contrast, the axonal network is affected and abnormal correlations are found in the case of Kir2.1 overexpression, although they do not resemble at all the phenotypes associated to the loss of *puc* (or JNK hyperactivation in RN2 cells). TNT overexpression had no effect on VNC axonal distribution as shown. We have now explored if diminishing Fas2 expression, a known consequence of the overactivation of the pathway in pioneer cells ((see our previous paper (Karkali et al, 2023)), could also affect motor activity coordination and we found that it does (Supplementary Figure 4). This is now discussed in the new version.

- Add Fas 2 VNC images next to correlation graphs (like in the previous paper) to make the interpretation more accessible to those who haven't seen something like this before.

Corresponding Fas 2 images are now added to the correlation graphs in Figure 3 and the Supplementary Figure 3 (old Supplementary Figure 2).

- In the discussion the authors are hypothesizing how the late-stage morphological changes and resting membrane potential may be interlinked and then go on to make the point that although this changes the property of the neuron, because these changes are not seen till the later stages they cannot be the cause of the lack of motor coordination in early stages.

However, the way it was stated is convoluted and it took me a few rereads to understand what they were saying.

We have rewritten the text in different sections to clarify this point.

- In the results on changes in motor coordination, the authors state that the phenotype is not due to loss of muscle integrity. There is no explanation as to how they came to that conclusion, so a statement on why they believe that would help.

The morphology of muscles is normal after the overexpression of the Hep^{CA} transgene in RN2 or MzVum cells in the current Figure 2a and b (this is re-stated in the new version of the manuscript). Further, we evaluated the integrity of the muscle pattern by immunostainings against muscle Myosin in *puc* mutants. We found that all muscles were kept in position and

look fairly normal despite the overall aberrant shape of *puc* mutants. This information is added as a New Supplementary Figure 2. Correlative Supplementary Figures have been renumbered accordingly.

- The authors claim that some phenotypes only show up in later stages, but only show the later time point where we can observe the phenotype. There is also no mention of testing earlier time points that did not show the phenotype.

Figure 7a to f and Supplementary Movie 6 show that the early and late phenotypes induced upon JNK signaling overactivation in RN2 cells are different (cell bodies positioning, axonal network organization, VNC length and motor coordination - we have now extended the muscle activity kymographs to early time points).

- Loss of JNK signaling also showed failure in VNC condensation in their previous paper, but only gain of signaling was used in this paper, whether that be disinhibition through puckered mutant or activation with HepCA. Given that both scenarios give the same loss in VNC condensation, they could include results for the loss of signaling as well since any difference in phenotypes may help inform the exploration of the mechanism in the future.

This assertion is not fully correct. BskDN affects the axonal network correlation and the nodes look slightly more distant than in WT embryos suggesting a lack of condensation (Figure 3, Karkali, 2023). Note that nodes distances are a proxy but do not provide unequivocal data on VNC length. On the other hand, direct measurements of the VNC length (Figure 5, Karkali, 2023) show a slight VNC hypercondensation. We have now revisited these data and confirmed them. Still, we agree that any information on the effect that JNK activity downregulation could have on motor activity coordination would be very relevant. Thus, we checked the motor phenotype caused by the loss of JNK activity in RN2 cells and found that this affects the last E stage of pre-hatching, which is prolonged, with some embryos occasionally hatching. Coordinated peristalsis were observed in stage C and the extended stage E (Supplementary Figure 4). This is now commented in the text.

Reviewer #3

The relationship between the architecture and functionality of the nervous system remains a topic of intense interest. Here, the authors investigate the relationship between JNK signaling and motor neuron development in the fruit fly embryo. The authors report that JNK signaling

is required for morphogenesis of the ventral nerve chord (VNC), in part by the action it exerts on a negative feedback loop mediated by the dual specificity phosphatase Puckered.

Specifically, the authors investigated how JNK signaling autonomously regulates neuronal electrophysiological properties. They showed that constitutively overexpressing JNK in motor neurons affected their ability to coordinate the development of motor circuitry required for hatching.

Although the authors analyze motor neuron activity, embryo movement, and mitochondrial transportation in motor neurons in which they specifically overexpress HepCA, it is unclear whether the functional deficit described above is the result of architectural changes or alteration of motor neuron function. In other words, the authors have not ruled out the possibility that the functional defects found in HepCA overexpressed motor neurons are due to structural defects in the mutants specifically, rather than to JNK signalling.

Indeed, we provide serial information supporting a direct link between JNK signaling, condensation/tissue organization defects (see Karkali et al, 2023) and the coordination of motor activities in late embryos. The disorganization of the neuronal landscape and the failure of tissue packaging, caused by high JNK activity in a specific set of pioneer motoneurons, at early time points of embryogenesis, impose structural restrictions that correlate with defects on muscle activity coordination. Intercellular signaling, intracellular transport or electrophysiological defects of these cells were ruled out (Kir2.1 and TNT experiments) and geometrical and structural constrains seem to be sufficient for disrupting muscle contraction patterns. The molecular mechanisms in part responsible for the neuronal cell adhesion changes associated to the VNC's structural organization phenotypes described are reported in our previous Nat Comm paper. We also present new data showing that diminishing Fas2 expression, a known consequence of overactivating the JNK pathway (Karkali et al, 2023), also disrupts motor coordination (Supplementary Figure 4).

Some of the data in the manuscript is unconvincing, and most of the experiments are not well quantified.

All analyses performed are totally trustable and any previously missing statistical information on their support is provided as specified below:

Specific Points:

1. In general, most of the experiments are not well quantified. Although the authors show some statistics, most of the time they do not clearly indicate the number of animals or sample sizes. For example, in Figure 1, the authors describe the contraction phenotypes by Stages A to E in the main text and show the evolution of the contraction waves graphically in Supplemental Figure 1. However, the authors provide no details on how the features they propose contribute to the different stages of contraction, for example, by using the features for segmentation of Stages A to E.

The detailed analyses of the different phases of locomotor activities in the embryo have been already determined by (Crisp et al, 2008). We confirmed all the features (behaviors) previously identified (Figure 1 and Supplementary Figure 1). It seemed redundant to timely and quantitatively analyze the different stages of motor activity maturation, thoroughly analyzed in this previous report.

Similarly, although the mutant phenotype is clear, the authors present little if any quantitative data. For example, they could provide the average duration of each stage, as well as the number of samples tested. Without sample sizes, it is difficult for any reader to assess the statistical significance of the findings. (For example, how many samples were tested in Figure 1C and D, Figures 2, 3, 4, etc.?)

We thank the reviewer for pointing us the need for a detailed quantification of our results in support of a clearer description of the mutant phenotypes. Although precisely staged animals were employed there was an unavoidable intrinsic variation in the length of the different stages of maturation (see also Crisp et al, 2008), and this variability increased in mutants. The kymographs displayed come from representative animals, and they point to the permanent arrest observed at discrete stages in the different experimental conditions. All the information on number of animals, samples sizes, the statistical tests performed and p-values for all data are now incorporated in the figures themselves, the figures legends and in the Source Data table.

2. Most of the experiments involve overexpression of HepCA, Does RNAi knockdown of *puc* in motor neurons RN2 or CQ2 similarly show HepCA overexpression, electrophysiological phenotypes, and/or mitochondrial transportation in motor neurons?

In the complete absence of *puc* (*puc^{E69}*), when the axonal network organization of embryos is altered and their VNC fails to condense, the progression of motor patterns is interrupted and embryos remain indefinitely in an immature stage (Figure 1). This is also the case when Hep^{CA} is overexpressed in RN2 cells (Figure 2). Any information regarding electrophysiological

phenotypes and/or mitochondrial transport upon silencing *puc* in motor neurons by RNAi would be redundant to the information obtained from overexpressing Hep^{CA}. Further, the penetrance and expressivity of *puc* RNAi phenotypes in the embryo is notoriously mild (tested with multiple Gal4 lines), as is often found for many RNAi transgenes.

3. When Kir2.1 is expressed in motor neurons RN2 or CQ2, is the morphology of motor neurons affected (Supplemental Figure 3). Please quantify how often each phenotype occurred.

The penetrance of the altered axonal and dendritic arborizations of RN2 and CQ2 cells described is very high (higher than 80 % in both cases). The actual figures (and sample sizes) are now included in the figure legend and the Source Data file. We present in the Supplementary Figure 5 (old Supplementary Figure 3) mild and strong phenotypes images to illustrate the variability in expressivity for these conditions.

4. Figure 4: Please indicate how many animals were used, as well as the number of mitochondria-positive dots. Also, please indicate the sample size. (For example, is N = 1? If so, is the average and bar necessary?). Furthermore, please indicate the type of statistical tests that were used to obtain p-values that show differences between genotypes (for example, in panels c to h).

Multiple animals and a high number of mitochondria were tracked. Missing statistical parameters (number of animals, sample sizes, Standard Deviations and tests employed) have now been incorporated in a new version of Figure 3 and also in the Supplementary Figure 6 (old Supplementary Figure 4).

5. I am wondering what is the mitochondrial transport phenotype upon RNAi knockdown of Jnk signaling in RN2?

Loss of JNK signaling by overexpression of BskDN in RN2 cells (and MzVum) introduces correlation defects but has a mild effect on VNC condensation (Karkali et al, 2023 Figures 3 b and e and Figure 5 k). The loss of JNK activity in RN2 cells does not produce any motor coordination defects and many embryos hatch (new data in Supplementary Figure 4). This is now commented in the text. Although potentially interesting, any information on the effect of JNK loss of function conditions, in which embryos manage to coordinate motor activities, on mitochondrial transport would not impact manuscript's conclusions and message.

Figure 6 c - f: It is difficult to see how p-value of < 0.001 can be obtained with a sample size of 4 with the substantial overlap of standard deviations between genotypes. Please confirm the accuracy of this statistical analysis

The sample size of 4 referred to the number of animals employed. The number of axons and dendrites analyzed was much higher. The numerical details are now provided in the Figure legend and the Source Data file. We confirmed the accuracy of the statistical analyses.

Minor points

1. Line109: Please explain what U1 – U5 motor neurons refer to.

The U neurons are a group of five per hemisegment eve-positive motor neurons. U1 to U5 are targeted by the CQ2-Gal4. We have added an appropriate reference in the text.

2. The figures could be improved by adding some legends. For example, in Figure 4a – h, it would be helpful to provide legends for the color code and genotype (for example, gray bar: control; green bar: HepCA > RN2; etc.).

Following the reviewer advice, we added legends to Figures 4, 5, 6 and Supplementary Figure 6 as it was suggested.

REVIEWERS' COMMENTS

Reviewer #1 (Remarks to the Author):

The paper titled “Puckered in pioneer neurons coordinates the motor activity of the *Drosophila* embryo” provides novel insights into the role of June Kinase (JNK) signaling pathway in pioneer neurons during embryonic development.

Puckered (Puc), a phosphatase, plays an important role in regulating the June Kinase (JNK) pathway and its activity in puc expressing neurons via a negative feedback loop. The findings of the study elucidate that overexpressing a constitutively active form of JNKK (HepCA) (and thus the loss of puc) in subsets (but not all) of the puc+ neurons leads to an increase in JNK signaling, thereby disrupting the vnc condensation and motor coordination. The fact that loss of puc in puc+ RN2 and VUM pioneer neurons (but not in puc+ CQs neurons) leads to the vnc condensation and motor coordination is interesting. Of particular interest, the elevated JNK signaling should occur at early embryonic stages to disrupt vnc condensation and motor coordination. Overactivation of JNK at late embryonic stages leads to dendrite and axon morphology defects, cell body mis-localization, membrane hyperpolarization and anomalies in mitochondrial transport. These effects are then confirmed to be directly from increased JNK signaling because functional perturbation (using TNT or Kir2.1) of pioneer neurons did not affect vnc condensation or motor coordination. It is then implicated that most of these phenotypes could contribute to JNK's role in wiring optimization.

The authors had previously shown that overactivating the JNK pathway leads to diminished Fas2 expression (Karkali et al, 2023). The revised manuscript also contains new data showing that Fas2 mutation also disrupts motor coordination, phenocopying the defects seen when JNK pathway is hyperactivated.

The authors have addressed the majority of concerns I raised in my initial review. Therefore, I believe the revised manuscript is now suitable for publication in Nature Communications.

A minor issue:

In Figure 7, the stages shown in the panels do not match the legend and main text. Kymograph data is labeled as panel g in the figure but as f and g in the legend.

Reviewer #2 (Remarks to the Author):

Authors answers most of my suggestions clearly. I recommend to publish this work.

Reviewer #3 (Remarks to the Author):

This paper grapples with a long standing and fascinating problem in developmental neurobiology: How does development of architectural features in the nervous system sculpt function and vice versa? This question is quite difficult to address systematically, but the authors have chosen what is arguably one of the best model systems (*Drosophila* embryonic ventral nerve cord) in which to address the problem. Their approach is sound and based on tried and tested approaches. Their data are convincing and align well with the claims made. The second round of revisions have addressed the first round of reviewer comments and added in quantitation where appropriate and necessary.

The number and diversity of genetic manipulations and approaches used in this publication is appropriate for a single paper and very clearly make this more than a purely descriptive study. The genetic manipulations and range of live imaging, anatomical and electrophysiological analyses used provide solid evidence for the role of JNK signalling in coordinating an array of functional and architectural features. Overall, this work on pioneer neurons is pioneering in the field. Taken together with their previous work and the work of others, this research sets the stage for, and lights the way for, future studies in invertebrates and in vertebrates.